# Improvement of Electromagnetic Side-Channel Information Measurement Platform

**DOI:** 10.3390/s23062917

**Published:** 2023-03-08

**Authors:** Shih-Yi Yuan, Wei-Sheng Liu

**Affiliations:** Department of Communication Engineering, Feng Chia University, Taichung 40724, Taiwan

**Keywords:** EMR measurement platform, EMR side-channel effects, NN model dataset building

## Abstract

Research has shown that when a microcontroller (MCU) is powered up, the emitted electromagnetic radiation (EMR) patterns are different depending on the executed instructions. This becomes a security concern for embedded systems or the Internet of Things. Currently, the accuracy of EMR pattern recognition is low. Thus, a better understanding of such issues should be conducted. In this paper, a new platform is proposed to improve EMR measurement and pattern recognition. The improvements include more seamless hardware and software interaction, higher automation control, higher sampling rate, and fewer positional displacement alignments. This new platform improves the performance of previously proposed architecture and methodology and only focuses on the platform part improvements, while the other parts remain the same. The new platform can measure EMR patterns for neural network (NN) analysis. It also improves the measurement flexibility from simple MCUs to field programmable gate array intellectual properties (FPGA-IPs). In this paper, two DUTs (one MCU and one FPGA-MCU-IP) are tested. Under the same data acquisition and data processing procedures with similar NN architectures, the top1 EMR identification accuracy of MCU is improved. The EMR identification of FPGA-IP is the first to be identified to the authors’ knowledge. Thus, the proposed method can be applied to different embedded system architectures for system-level security verification. This study can improve the knowledge of the relationships between EMR pattern recognitions and embedded system security issues.

## 1. Introduction

The numbers and applications of embedded devices and/or the Internet of Things (IoT) are prevailing [1]. Therefore, the security issues of such devices have drawn much attention recently [2]. One such security threat is a side-channel attack (SC attack or SCA) [3]. SCAs exploit informative emissions from integrated circuits (ICs), which can be in the form of electromagnetic radiation (EMR), sound, light, power consumption, and/or timing information.

There are SCA studies related to special-purpose embedded systems [4,5,6,7]. However, there are relatively few discussions about the EMR of SCAs on general-purpose microcontrollers (MCUs). It has been observed [8,9,10] that when different machine instructions are executed in an MCU, the EMR patterns emitted by the MCU are slightly different. This means there is some execution sequence information leaked through the EMR SC. We call it an “EM-leak” pattern corresponding to each instruction execution EMR SC leakage in this paper. Although it is hard for humans to identify such EM-leak patterns (Figure 1), numerical analysis [11] or neural networks (NNs) can be applied to find them.

Every program content and its execution sequence of the same MCU are then identified through the mapping set [10]. All the necessary instructions can be mapped to these patterns using a specific characterization program (test program) that contains all the EM-leak patterns. The mapping set of “(EM-leak-1, instruction-1),…, (EM-leak-n, instruction-n)” can be constructed once all of the mappings have been found. 

The efforts of the work series ([9,10,12,13,14]) are generally to build two frameworks: one for data acquisition (methodologies, techniques, platforms, etc.) and the other for data analysis (methodologies, NN model and architectures, generalization of different MCU architecture measurement foundations, etc.) on single-type MCU EM-leak problems. However, this paper focuses on the data acquisition platform improvements only, which mostly belong to the data acquisition part of the work series. As the performance comparisons are to be made meaningfully from the whole system (fair comparisons), the proposed platform comparisons must be described based on the integration of the two frameworks with other parts of the two frameworks remaining the same (an ablation study). We can only choose [10] for comparison because it is state of the art to the best of the authors’ knowledge. 

The current EM-leak studies generally focus on specific functional chips (DSPs, encryption/decryption chips, among others) [4,5,6,7], while the work series focuses on general-purpose MCU EM-leak issues. In addition, the MCU-EM-leak issues are much harder compared to the specific function chip EM-leak problems. This is because this issue can be generalized to a binary classification problem, which is a special case of this research (a multiple classification problem).

The platform is not used for the encryption process. The purpose of the measurement platform is to gather EMR from MCUs in order to reduce the possibility of SCAs. However, other research on the EMR information leakage of specific chips can benefit from this research.

Each measurement in [10] has 1,000,000 samples (sample points) of the EMR signal taken every 10 ms. The EMR data sampling rate is 100 samples within µs. The 10 ms EMR measurement period includes approximately 256 machine instructions executed during the period. The EM-leak patterns can be mapped to these instructions using the techniques proposed by [10]. The data processing procedures can be briefly described as follows: The EMR data are first divided into fragments of EM leaks. By mapping each EM leak into each instruction (Figure 1), the mapped instruction sequence can be determined. This means the context of any instruction execution sequence can be identified after the mapping set is determined. The mapping procedure will be described in Section 2.

Although [10] can be used for a preliminary EMR SCA, there are still difficulties with detail analysis and for different types of embedded systems. In [10], every EM leak contains 273 samples, and the identification is performed by an NN. The instruction can be roughly identified from the EM leak by the NN. Although it is better than random guessing (about 6.25% accuracy) for six times improvement, the analysis accuracy is still very low (top1 accuracy ≅ 37.78%). However, it is the best result to the authors’ knowledge. 

The measurement may be the source of the problem. The delicate fluctuation characteristics of the waveform might not be captured because each EM leak only has 273 samples. This could result in the NN learning too few features, which would lower analysis precision. The low resolution also hinders the method applied to other types of MCU architectures, such as FPGA-based softcore MCUs. The accuracy of the final data analysis will be impacted by the fact that every movement of the measurement platform will result in a small positional displacement and subtle data measurement distortion.

Due to the above problems, this paper proposes an automated EM-leak measurement method. It uses a Cartesian robot as the measurement platform, which not only reduces the motion displacement but also improves the sampling rate using automation. These improvements can increase EM-leak features used in NN analysis.

In the previous study [10], the device under test (DUT) was limited to general-purpose MCUs (the DUT is the microchip dsPIC33EP512MC202 MCU). As the original platform can only be moved and measured manually, it is only suitable for ASIC-like MCU EM-leak analysis. For more flexible circuit designs, such as FPGAs, the platform cannot find any features good enough for EM-leak pattern identification. It may be that, at low resolutions, the pattern features cannot be effectively identified, and the mapping fails.

The proposed automation platform in this paper increases the sampling rate up to 1428 samples within µs (14X higher than [10]) and reduces motion displacement errors, thereby improving signal analysis and mapping capabilities. Thus, the proposed platform extends the EM-leak feature analysis to FPGA-IPs as a new DUT and increases the application field.

This paper adopts the similar NN analysis method as [10] and focuses on improving an integrated software and hardware platform to improve measurement stability, automatic controllability, and measurement repeatability, and reduce measurement manpower. 

## 2. Electromagnetic Radiation (EMR) Side-Channel Analysis (SCA) through a Neural Network (NN)

### 2.1. Artificial Neural Network (NN) and the Measurement Platform

An artificial NN is a computational model that imitates the structure and function of neurons in human brains. It performs calculations through a large number of artificial neuron connections. During the process, an NN will obtain data features, give weights, add up the features through the weights, and finally determine the final answer. During the process, the numerical training and learning of the weights will be changed so that the calculated results can be close to the optimal results. NNs must be trained first and need many labeled data. There are many ways to obtain the data. For example, some research is based on previously established datasets [15,16], and some problems can be gathered using web crawlers [17].

For the EM-leak mapping problem, the dataset can only be measured using a specially designed platform. If the data features are affected not only by the problem itself but by the data collection process, such as the number of samples, the positional displacements, the stability of the repeated sampling, etc., it may lead to NN feature distortion and affect the EM-leak mapping quality. If the size of the training set is not large enough or the quality of data is unstable, the training quality can be poor. Therefore, training a better NN architecture requires both better quality and quantity of the training set of the EM-leak mapping problem. Thus, an automated and stable measurement platform is required.

### 2.2. Different EMR Patterns of Machine Instructions

In previous studies [8,9,10], it was confirmed that the EMR patterns emitted by MCUs are different when executing different instructions. However, the difference is subtle, and it is quite difficult to differentiate using human vision. In previous studies, convolutional NN (CNN) [8] and multilayer perceptron (MLP) [10] are used for EM-leak identification, but the accuracies are still low. Among them, the model with the highest mapping accuracy is the MLP method [10], which combines time domain (Figure 2) and frequency domain (Figure 3) EM-leak feature extraction NN training. The model architecture is shown in Figure 4. The model architecture in Figure 4 is a dense neural network with seven dense and dropout layers; its inputs are time domain and frequency domain. The training results are shown in Figure 5. The top1 testing accuracy rate is 37.78%, and the top5 accuracy is 90.75%. Although it is already six times higher than random guessing, the top1 accuracy is still low, and there is still a lot of room for improvement. 

### 2.3. The Two Devices under Test (DUTs)

#### 2.3.1. dsPIC MCU [18]

One of the DUT targets is an MCU (microchip dsPIC33EP512MC202). It is a two-stage pipeline MCU. The printed circuit board (PCB) is designed by the authors. Figure 6 shows the appearance of the MCU. The test program is designed by an assembly program and is programmed (burnt) using the MPLAB X IDE tool. The test assembly program mainly executes 18 target instructions following a fixed time delay and loops back indefinitely. The EMR signal fragmentation is cooperated by a system clock signal and an IOMark [14]. The IOMark will be briefly described below.

The test program instructions are shown in Table 1. The dsPIC33EP512MC202 is a two-stage pipelined MCU. It means the MCU executes one instruction; in the meantime, a second instruction is taken from the external memory (fetched) and decoded. As shown in Table 2, there are two instructions within the same clock cycle. For example, in clock cycle 2, the inst.1 is executed by the MCU, and inst.2 is fetched by it. In this paper, it is represented as “Inst(N) + Inst(N + 1)”. All the execution patterns considering the two-stage pipeline architecture are shown in 0. As the “MOV + INC” is executed in the loop twice (two 9s in 0), there are only 16 EM-leak patterns. A brief description of each instruction is shown in Table 3. And the EM-leak patterns are shown in Table 4.

#### 2.3.2. FPGA-Based CPU

Another testing target is a simple 32-bit processor called CPU0 [19]. The circuit is described in Verilog. The Verilog CPU0 [19] DUT is implemented using the Quartus FPGA design tool [20]. After the Verilog code is verified, the CPU0 IP (an FPGA-based softcore IP) is burnt into an FPGA (Intel Cyclone IV EP4CE6E22C8N [21]) to execute its instructions.

A similar test program (Table 5) as Table 1 is adapted for CPU0. In addition, the EM-leak patterns can be obtained and analyzed similarly. A brief description of each instruction is shown in Table 6. As CPU0 has no pipeline architecture, there is no need to consider the pipeline EMR effect such as Table 2.

### 2.4. Automated Measurement Platform

Robotic arms are now practical in many applications [22,23]. The robotic arms can be divided into many types (for example, the Cartesian robotic arm type, the gantry arm type, the cylindrical arm type, the spherical arm type, and the selective compliance assembly robot arm (SCARA) type, among others [24]). Types of robotic arms can be used for different applications. The assistance of a robotic arm can increase productivity and minimize errors in high-repetition and high-precision work. The robotic arm can be used stably in repetitive tasks accurately.

In this paper, the platform is required to perform a large number of EMR measurements with high repetitive stability, low positional displacement, and high DUT flexibility. In conclusion, the gantry-type robotic arm is chosen for the measurement platform. The gantry arm is an improved version of the Cartesian arm with two X axes instead of one. Each axis has a stepper motor to control the direction and distance of movements. Additional control axes (sometimes with additional Y and Z axes) allow the robot to handle faster movement with high localization accuracy [25].

Localization accuracy is the maximum positional displacement errors of repeated motions to the same position, which can be identified by analyzing the gear ratio and the minimal step angle of the stepper motor. In this paper, when the load is under 30 g (the EM probe used in this paper weighs 20 g) and without an optical ruler for further displacement improvement, the accuracy of the proposed platform can be limited to ±10 µm. Compared with [10], with the original displacement error being about ±200 µm, the localization accuracy is about 20 times improved.

The proposed system uses a personal computer (PC) to drive the robotic arm and interacts with a real-time oscilloscope at the same time. After moving the EMR probe to a pre-determined position, the PC controls the oscilloscope to measure and record the EMR patterns. The schematic is shown in Figure 7. The picture of Figure 7 is shown in Figure 8. The detailed control flow is shown in Figure 9. The oscilloscope and computer are connected by an Ethernet cable controlled by MATLAB. The bidirectional communication is controlled by the virtual instrument software architecture (VISA) protocol. The Modbus connection [26] allows the computer to control the robotic arm. Both protocols can interact with the MATLAB control program.

Figure 10 is a flowchart of DUT execution, including Testkernel and GI. Figure 11 is the raw EMR waveform (red line) obtained by the platform. The blue line is the IOMark signal. The period of IOMark in high voltage is the EMR signals of a TestKernel. Figure 11 also shows the EMR corresponding to five TestKernels (TestKernelEMR), where the indigo line is the system clock, the blue line indicates TestKernelEMR/GI boundaries, and the red line is the EMR. Figure 12 shows one TestKernelEMR. As each TestKernelEMR contains not only the TestKernel instructions but also two IOMark (set high and set low) instructions, the final EM leaks of each TestKernelEMR are further identified and fragmented using system clocks. Each EM leak corresponding to the specific instruction execution sequence pattern is identified using a clock, as shown in Figure 12. The EM-leak patterns are shown in Figure 12. For example, the EMR in the black box is the EM leak for pattern 1. According to 0, the corresponding execution sequence patterns of the EM-leak pattern 1 is “Mov + Mov”. The final EM leaks corresponding to specific instructions can be obtained, similar to Figure 1.

### 2.5. Data Normalization

When collecting the EM leaks, the proposed platform can inevitably be affected by noises. These fluctuations may come from clock jitter or external noise. In addition, they may affect the timing period windows of the corresponding EM leaks, and a normalization algorithm similar to [13] is developed for such fluctuation normalization.

The algorithm first converts the IOMark signal to the frequency domain using discrete Fourier transform (DFT) to find the main frequency using the highest power response. It is used to determine the largest possible period at a particular window period. After the frequency is determined, the signal is transformed back to the time domain through inverse DFT (IDFT). The main frequency becomes the ideal switching period determined by the algorithm to eliminate the noise influence of the IOMark. The normalization concept is shown in Figure 13. The same technique is also applied to find accurate clock windows inside an IOMark period.

## 3. The Proposed Platform Implementation

### 3.1. Platform Setups and Procedures

The specification of the controlling platform PC is Intel I5-4670 with 16 G DDR4 SDRAM equipped with VISA and Modbus protocol interfaces. The controlling program is designed by MATLAB R2020b. Through the PC, the measurement platform controls a real-time oscilloscope (R&S^®^ RTE1104, bandwidth is 200 MHz to 2 GHz, the sample rate is 5 G sample/s), an EMR probe (RF-K 7-4, the resolution is 5 mm), and the robotic arm. 

The experiment is divided into 3 main steps. The first step is to measure all the signals (EMR, IOMark, and system clock) from the DUT through the proposed automatic measurement platform. The signals are transmitted to the computer for data processing. Second, the IOMark and clock signals are normalized. The EMR signals are then identified through the normalized IOMark and clock to EM leaks. Each EM leak is labeled with the corresponding instruction according to the IOMark and clock signals. The EM leaks and their labels are extracted and stored as training data. The third step is the NN training using the stored data. 

After the NN training procedure, the NN model is tested with newly measured EM-leak signals from different test program(s) to predict the instruction execution content and sequence. The accuracies are collected to determine the prediction quality of the proposed platform.

### 3.2. Signal Acquiring and Processing

The data acquisition process is shown in Figure 9. Before the measurement, the number of measurements (measurement count) and the number of positions per measurement are given. After the initialization of the robotic arm and the oscilloscope, the platform starts the measurement. Robotic arm takes 0.5 s to move from point to point, and the distance between two points is 2mm. Robotic arm movement path is meandering. Once the robotic arm reaches the specified position, the oscilloscope starts acquiring data and sending it back to the PC. The processes are repeated until the measurement count is reached. Each measurement is 1,000,000 samples within 70 µs in this paper, which is about 14.28 times higher compared to [10] (1,000,000 samples within 10 ms).

As the sample rate is 14.28X higher than [10], the time required for measurement is expected to be increased correspondingly. In [10], nearly 16 IOMarks can be obtained from one measurement. In the proposed platform, only 5 IOMarks can be transmitted per measurement due to the higher sample rate. However, the new proposed platform can reduce the required manpower by following two improvements. 

First, we improve the robotic arm measurement control algorithm. Due to the high variation delays of the old robotic arm actuation time, paper [10] needs a long GI to maintain the measurement synchronization. In this paper, we reduce the unnecessary EMR measurement GIs between IOMarks. This is because the new proposed platform uses a better robotic arm actuation mechanism that can cooperate better software driver algorithms, faster response times, and a higher driving current. Thus, the authors can keep the EM-leak data with fewer GI delays. In [10], due to the semi-automation that needs human intervention, the ratio of the training data to the GI data is 1:12. There is obviously too much time wasted transmitting the GI data. The ratio is reduced to 1:2 in this paper because of the improvements above and smoother HW-SW co-design. 

Second, the proposed platform is more automatic and stable. The platform can be operated for 24 h. Thus, the time required for human intervention is significantly reduced. In [10], manpower was required to observe the entire measurement period to avoid measurement errors. In the new proposed platform, it is only needed to initialize the platform (setting the measurement parameters) before the measurement. After initialization, the platform can automatically perform stable measurements until completion.

When the probe moves to a new position, it collects all the signals there (Figure 11). Higher resolution means more features can be extracted at the cost of longer data transmission and data processing time. As the amount of data that the oscilloscope transmits each time is fixed, higher resolution IOMark means that less IOMark can be collected within one oscilloscope transmission period. In this paper, only 5 IOMarks per measure are transmitted. Currently, the EM-leak resolution is 14.28 times higher than [10]. The resolution can be further increased if necessary. 

The previous study [10] shows that the best NN model needs features containing both the time domain and frequency domain EM-leak information. Thus, in this paper, the NN models and training features are kept the same as [10] for the platform comparisons.

### 3.3. NN Model Training

This experiment uses an MLP model similar to [10] (Figure 14). As the input size is larger than [10], the model should be adapted. However, all procedures for the data analysis are similar to [10] for fair platform comparisons.

## 4. Experiment Results

The DUTs are dsPIC and FPGA CPU0 and are described in Section 2.3. The proposed platform is described in Section 2.4. The platform setups and experiment procedures are described in Section 3.1. In addition, the signal measurement and processing are described in Section 3.2.

### 4.1. Measurement Result Comparisons

The detailed comparisons are shown in Table 7. Columns 1 to 3 in Table 7 are the comparisons of [10], the platform without (w/o) modified GI, and the platform with (w/) modified GI. 

The sample rate (row A), resolution ratio (row B), and displacement error (from ±200 µm to ±10 µm in Section 2.4) are better than [10]. In Section 3.2, we describe that the measurement time of one EM leak is longer than [10], but many improvements are made to reduce the measurement time and manpower.

The detailed analysis of the reductions is shown in the following calculations (Table 7, rows C–J). The oscilloscope initialization is 120 s before transmission. Each transfer takes a fixed 60 s period, and fixed 1,000,000 EMR samples per transfer are taken. The length of each TestKernelEMR (Figure 11) is averagely 5 µs (which will be used in the IM formulation). If the resolution is increased, the total transmission time (Section 3.2) grows correspondingly. Thanks to our measurement platform, the GI (Section 3.2) can be reduced. As a result, the data transmission time is increased not 14.28X but 7.86X and eventually reduced to 3.16X. The data transmission time is shown in row H of Table 7. The detailed comparisons are shown in rows A–J.

The sum of TestKernelEMR time (5 µs) and GI time is called one fully captured waveform (row D). Row E shows the ratio of rows D/A, which means how many samples per fully captured waveform. It can be seen that due to the lower resolution of [10] (row A), fewer samples are required. The large increment in samples at row E, column 2 (65,000 << 928,571) is due to the resolution increment (Δt, row A). However, the number of samples required can be greatly reduced by reducing the GI (row E, column 2 vs. column 3, 928,571 > 214,285). As the data from the oscilloscope transmitted are fixed per data transmission, the time it takes is constant (60 s). Rows A–E above can be used to calculate the number of IOMarks per oscilloscope transmission (row F). However, it is necessary to calculate how many IOMarks each oscilloscope transmits first. The calculation is how many complete waveforms (row E) are included in 1,000,000 samples. We also need to consider the case where the captured waveform is not full, but the last IOMark is completely included. Here, the authors define an “incomplete measure wave” (IM), which represents the status of whether or not the last IOMark (5 µs) is completely included in the transmission. If it is, the IM is 1. In addition, it means that the IOMark number per oscilloscope transmission can be increased by 1. The formulation of row F is as follows:$$\mathrm{row}\;F=[\frac{1,000,000}{\mathrm{row}E}]+\mathrm{IM},\;\mathrm{where}\;\mathrm{IM}\;=\{\begin{array}{cc} 1, & if\left[\frac{(\frac{1,000,000}{\mathrm{row}\;E}-\left[\frac{1,000,000}{\mathrm{row}E}\right])*\mathrm{row}D}{5\;µs}\right]>1\\ 0, & otherwise \end{array}$$

For example, in [10], 1,000,000/65,000 (1,000,000/row E, column 1) = 15.38, which gives 15 fully captured waveforms. In addition, the IM is 1. So the number of IOMarks is 15 + 1 = 16 (row F, column 1). It means 16 IOMarks are included within one data transmission period.

In the previous experiment [10], the number of training sets for each pattern was 1600. In other words, at least 1600 IOMarks need to be captured. If 1600 IOMarks using three methods (the three columns) are to be measured and compared, it can be concluded that they need 100, 800, and 320 transmissions, respectively (row G). Considering the initialization and the transmission time, the total measurement time (row H) can be calculated as (IOMark number/row F) * 60 s + 120 s = 6120, 48,120, and 19,320 s, respectively. In addition, the ratios of the measurement time comparisons are shown in row I. There is only 3.16 times transmission delay compared to 14.28 times which is a brutal force application to [10].

Considering the total manpower, the new platform needs human intervention at only the initialization time. Thus, row J shows the total manpower measuring 1600 IOMarks EMR time. The ratios are further reduced to 0.0196 both for columns 2 and 3 (from row J: 120/6120).

### 4.2. Analysis of Different DUTs

Through the measurement platform proposed in this paper, the EM leaks of the two DUTs (dsPIC33 and CPU0 FPGA-IP) can be derived (shown in Figure 15 and Figure 16). It is clear that the EM-leak waveforms are very different. In addition, different NN models should be built and trained through the same process described above.

### 4.3. The Accuracy Comparisons of NN Models

According to the traditional NN model performance evaluation, the top1 accuracy is to be evaluated. The training and verification top1 accuracies of dsPIC are shown in Figure 17. The same top1 accuracies of CPU0 FPGA-IP are shown in Figure 18.

The comparisons of the NN model accuracy of [10] and the proposed platform are shown in Table 8. From (a) and (c), we improve the accuracy of dsPIC by 1.46% compared to [10]. Although the accuracy is still not high enough, it is the highest top1 accuracy to the best of the authors’ knowledge. It is proved that high resolution can indeed improve accuracy.

The comparisons of another DUT (CPU0) are shown in (b) and (d). Table 8 shows the large accuracy improvement of the DUT (CPU0) (Section 2.3.2). In [10], the accuracy cannot be found effectively; the top1 accuracy is similar to random guessing in (b). Compared with [10], our accuracy has improved by almost 18 times (98.71%). As the EM-leak acquiring and NN model building procedures are similar, the drastic accuracy improvement may mainly come from the higher resolution for successful NN training. This observation also justifies that the higher sample rate may be essential for different DUTs.

The accuracy of CPU0 (d) also outperforms that of dsPIC MCU (c), which was not expected by the authors. We think the higher accuracy of CPU0 may be that the new platform’s high resolution facilitates FPGA-type IP EM-leak identification. Alternatively, the original dsPIC MCU is essentially too complex (two-stage pipelined MCU) for the current EM-leak identification algorithm: an EM-leak pattern containing two consecutive instructions that are executed at the same time inside an MCU is too complicated for the MLP-type NN model classification process. Therefore, although the improvement of resolution is beneficial to EM-leak analysis, the pipelined MCU is inherently more difficult to identify.

In a fair comparison, the platform is much improved from [10]. The improvements are non-trivial and heavily modified (for example, time-skew improvements, TestKernel identification, control automation for NN model dataset generation, HW/SW co-integration, and target chip extensions). All these improvements make high-resolution data acquisition and time efficiency possible. Many issues are discussed and compared based on [10] in Table 7.

## 5. Conclusions

The usage of embedded devices is increasing, and the security concerns of these devices are important. This paper optimizes the design of an electromagnetic side-channel leakage (EM-leak) measurement platform. 

To make accurate EM-leak measurements upon different DUTs (dsPIC MCU and FPGA-IP), a previously designed platform is improved in this paper. By modifying the original platform to the gantry-type robotic arm, the positional stability and repeatability of the measurements are improved. Using the synchronization signal normalization algorithm, EM-leak fragmentation accuracy is improved.

When data are collected automatedly, manual interventions and errors can be reduced. As the data measuring resolution increases, more EM-leak features can be extracted for neural network (NN) model training and verification. The improvements can help to build better EM-leak identification models and finally improve prediction accuracy.

From the authors’ viewpoint, the proposed platform indeed makes non-trivial improvements. Through the proposed platform, the sampling rate is increased by 14.28 times, and the resolution is increased by 14.28 times. Due to the GI reduction skills we introduced, the measurement time needed can be reduced from 14.28X to 3.16X. In addition, using the automation algorithm, the on-site manpower can be further reduced to 0.0196X.

When the resolution is increased, the top1 NN accuracy of the dsPIC is slightly improved by 1.46%. There is a new DUT in this paper: an FPGA-CPU-IP. To the authors’ best knowledge, this is the first report that the EM-leak measurement platform can be extended to other types of FPGA-CPU-IP DUTs. The accuracy is quite high compared to the previous platform; the top1 accuracy is 98.71%. In the previous study, a valid NN model could not even be built. Its accuracy even outperforms the dsPIC both in the original and new proposed platforms. 

We think the possible reason may be that the CPU0 is easier to be distinguished at higher resolutions. However, the dsPIC microcontroller is a two-stage pipelined MCU. When an instruction is executed, the EM-leak pattern contains two consecutive instructions that are executed at the same time inside the MCU, which complicates the model identification process. Therefore, although the improvement of resolution is beneficial to EM-leak analysis, the pipelined MCU is inherently more difficult to identify. As the NN model building method and the data processing procedures are the same, the observation justifies that a higher sample rate may be essential.

Through this paper, a newly proposed platform indeed improves the NN model accuracy on the EM-leak analysis problem with fewer positional errors, higher data acquisition resolution, better data process algorithms, and more control automation. The platform can be extended to different types of ICs for EM-leak studies. In the future, the authors will continue to improve the platform and data processing algorithms for better EM-leak understanding.

## Figures and Tables

**Figure 1 sensors-23-02917-f001:**
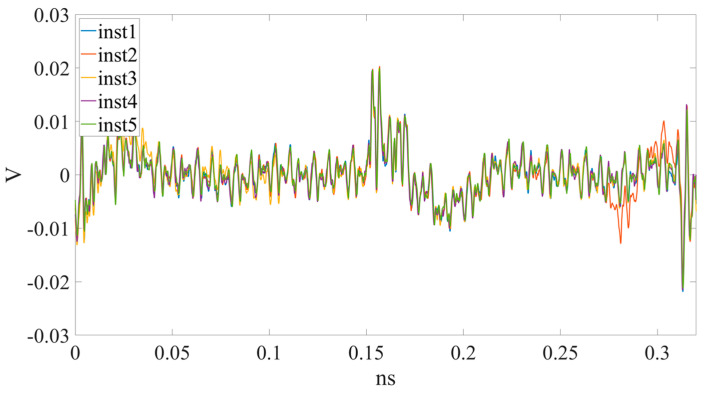
Different EM-leak signals of the FPGA-MCU-IP DUT corresponding to different instructions measured by the proposed platform.

**Figure 2 sensors-23-02917-f002:**
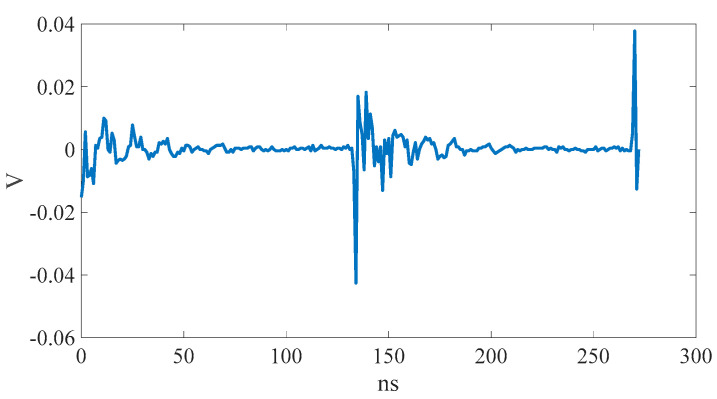
dsPIC33EP512MC202 time domain EM leak (adapted from [10]).

**Figure 3 sensors-23-02917-f003:**
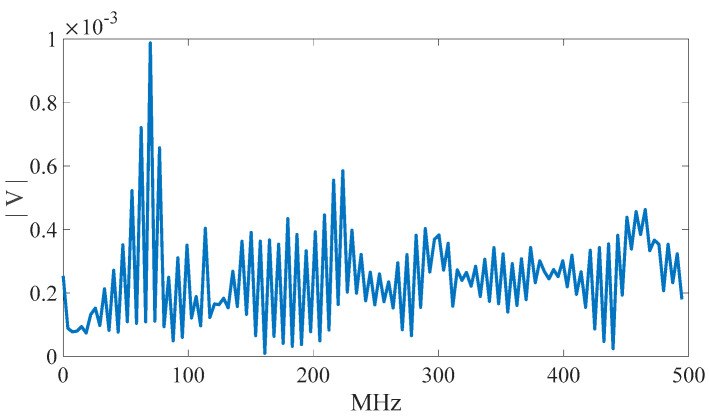
dsPIC33EP512MC202 frequency domain EM leak (adapted from [10]).

**Figure 4 sensors-23-02917-f004:**
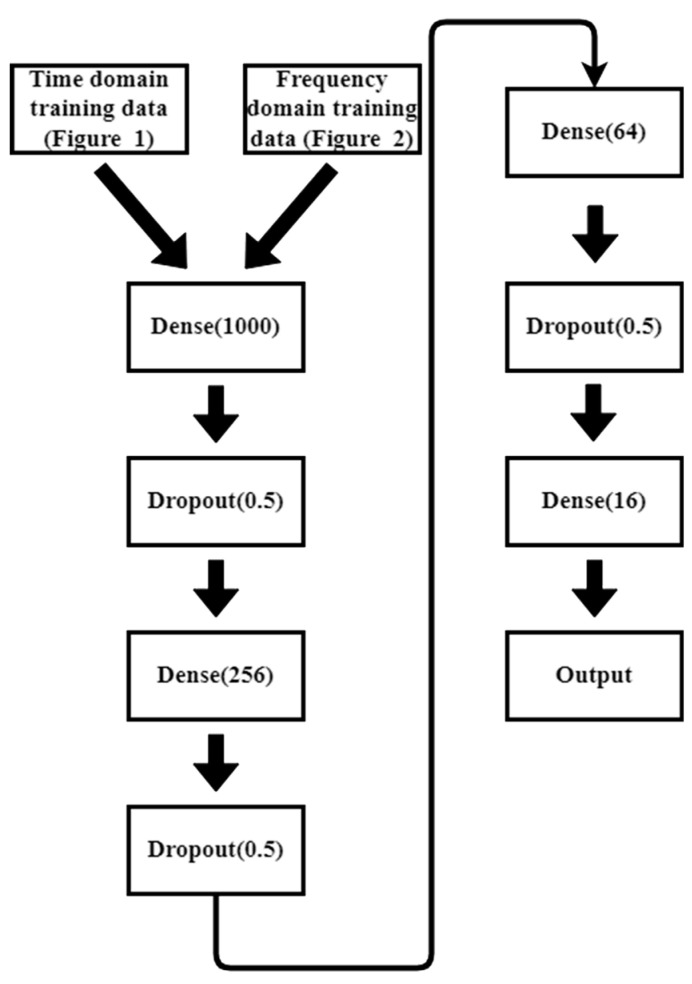
Time and frequency domain two-input NN model (adapted from [10]).

**Figure 5 sensors-23-02917-f005:**
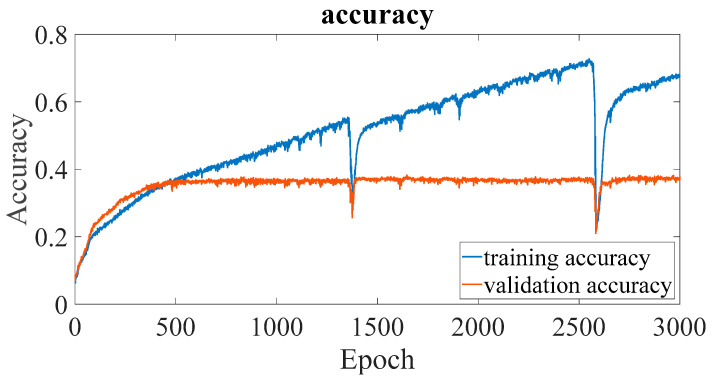
Training and validation accuracy of Figure 4 (adapted from [10]).

**Figure 6 sensors-23-02917-f006:**
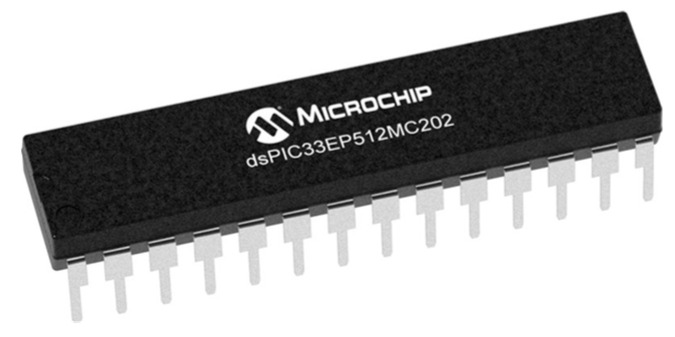
dsPIC33EP512MC202 [18].

**Figure 7 sensors-23-02917-f007:**
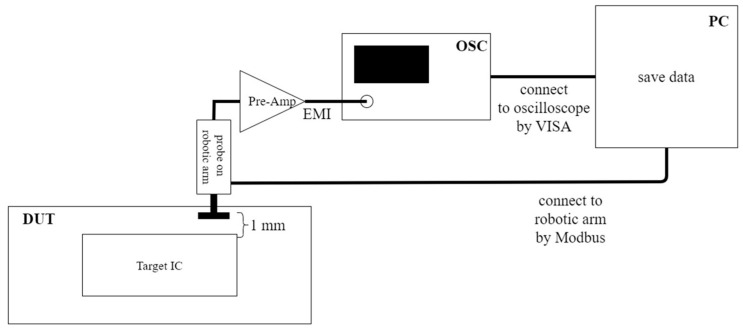
The schematic of the proposed measurement platform.

**Figure 8 sensors-23-02917-f008:**
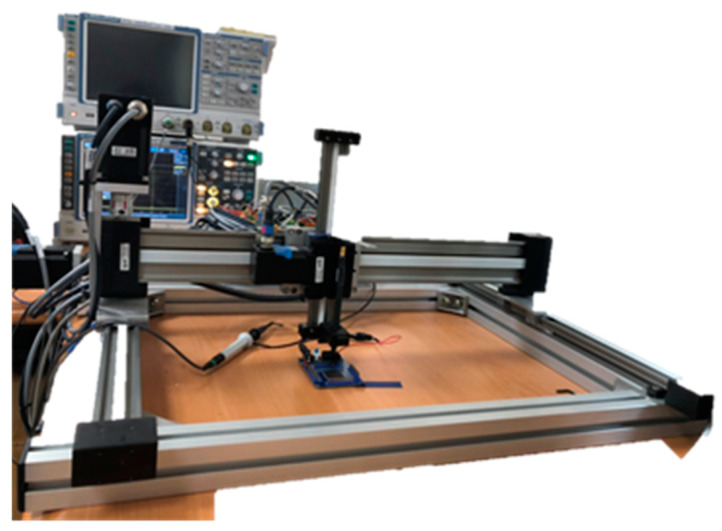
The platform of Figure 7.

**Figure 9 sensors-23-02917-f009:**
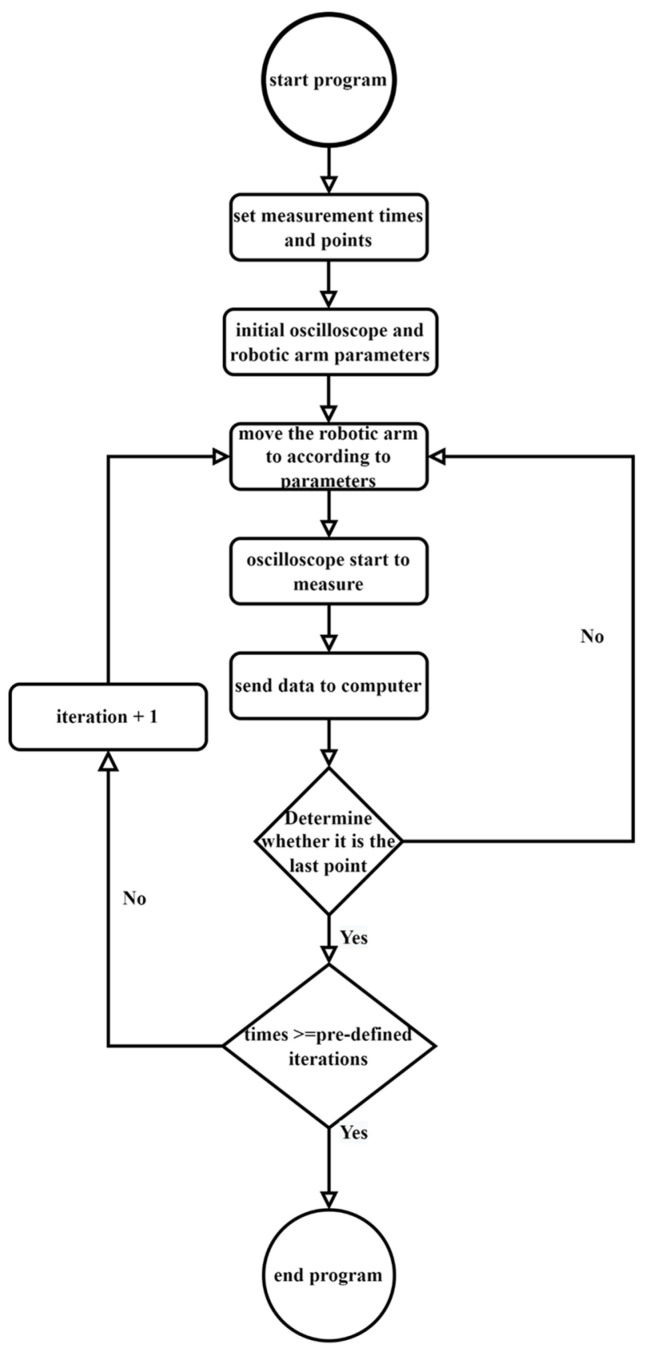
The control flow of Figure 7.

**Figure 10 sensors-23-02917-f010:**
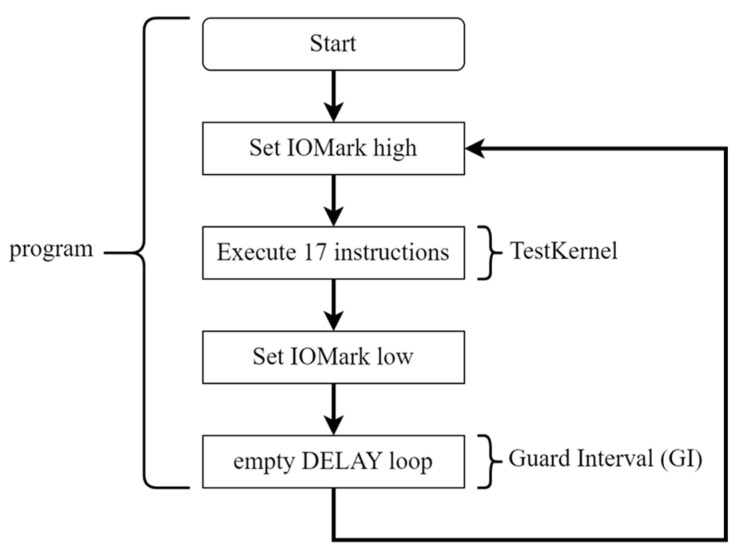
TestKernel flowchart.

**Figure 11 sensors-23-02917-f011:**
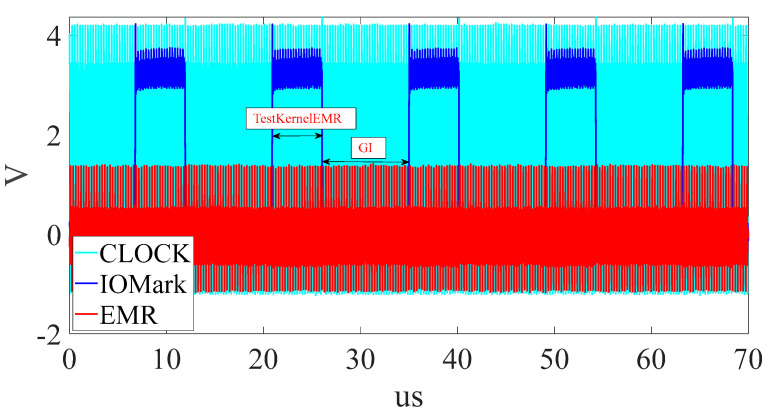
The measurement of the proposed platform.

**Figure 12 sensors-23-02917-f012:**
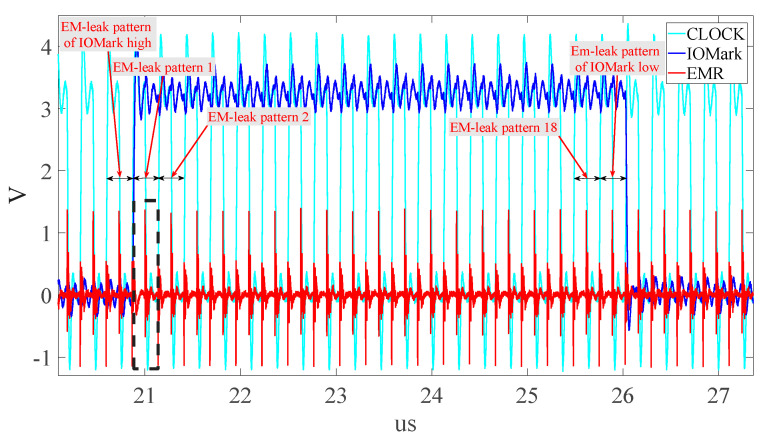
TestKernelEMR (EMR within IOMark) from Figure 11.

**Figure 13 sensors-23-02917-f013:**
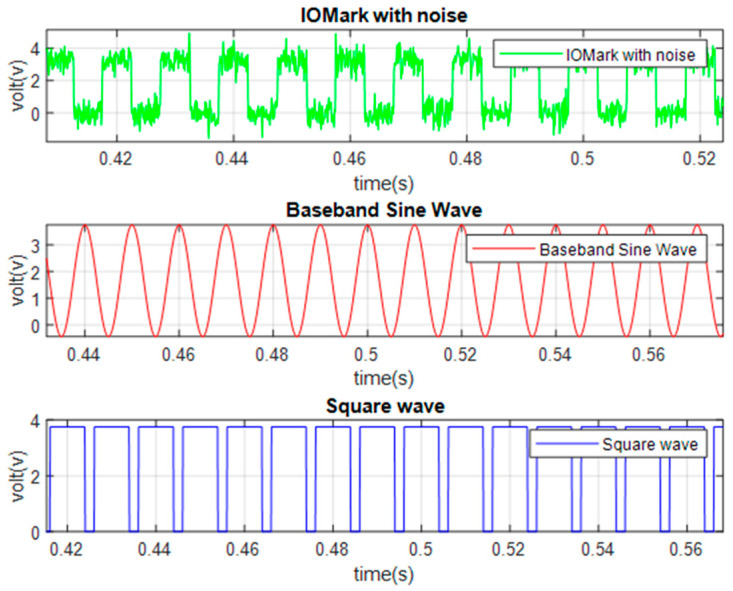
Clock signal normalization process (adapted from [13]).

**Figure 14 sensors-23-02917-f014:**
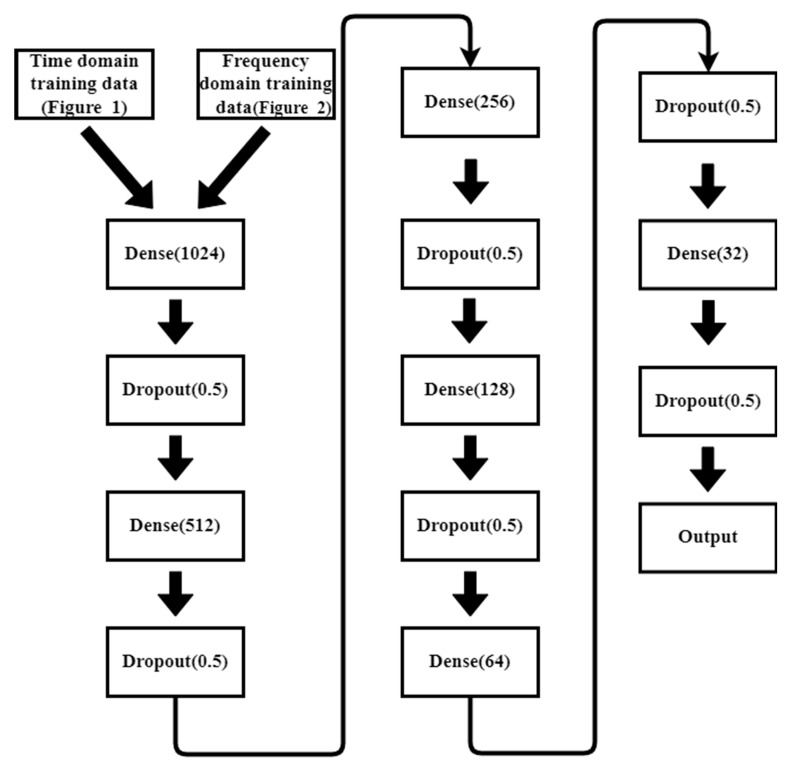
Multilayer perceptron (MLP) architecture used in this paper.

**Figure 15 sensors-23-02917-f015:**
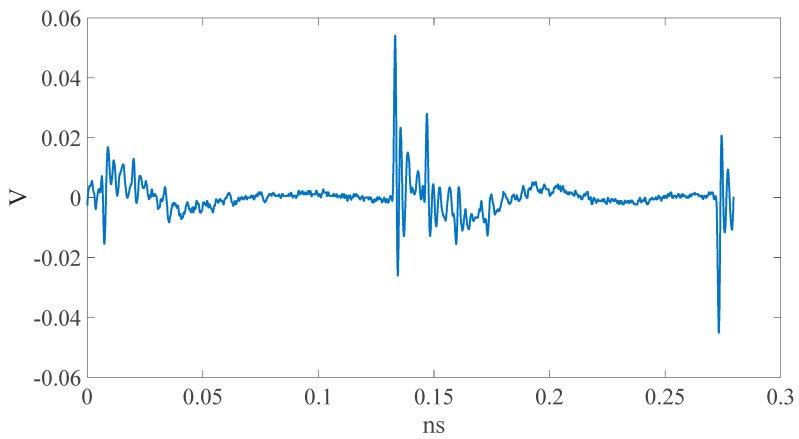
dsPIC33EP512MC202 EM leak.

**Figure 16 sensors-23-02917-f016:**
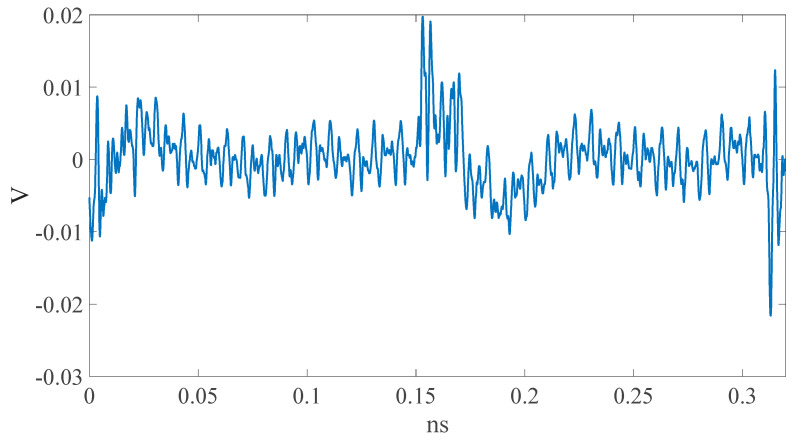
CPU0 EM leak.

**Figure 17 sensors-23-02917-f017:**
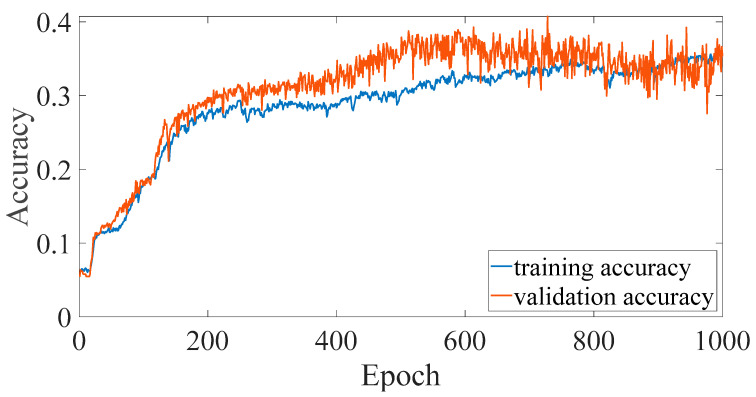
dsPIC top1 accuracy.

**Figure 18 sensors-23-02917-f018:**
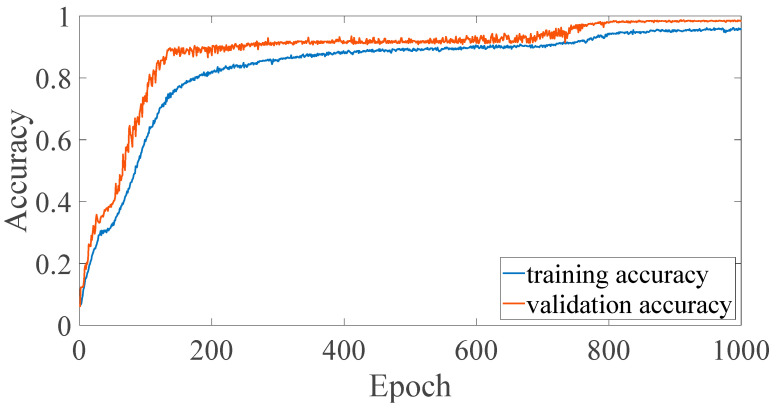
CPU0 top1 accuracy.

**Table 1 sensors-23-02917-t001:** dsPIC33EP512MC202 instruction table.

Mov	2. MOV	3. ADD	4. ADD
5. Nop	6. Nop	7. Inc	8. Inc
9. Mov	10. Inc	11. Nop	12. Mov
13. Inc	14. Add	15. Mov	16. Nop
17. Add	18. Bclr		

**Table 2 sensors-23-02917-t002:** Two-stage pipelines.

	Clock Cycle	1	2	3	4
Inst No.					
1		Fetch	Execute		
2			Fetch	Execute	
3				Fetch	Execute

**Table 3 sensors-23-02917-t003:** dsPIC33EP512MC202 instruction introduction.

Instructions	Description	Instructions	Description
Mov	Put value into a specified register	Add	Add two registers’ values
Inc	Increase the register’s value by 1	Nop	Do nothing
Bclr	Bit clear of file/Ws register		

**Table 4 sensors-23-02917-t004:** Execution sequence patterns of Table 1 (considering the DUT’s two-stage pipeline architecture).

1. Mov + Mov	2. Mov + Add	3. Add + Add	4. Add + Nop
5. Nop + Nop	6. Nop + Inc	7. Inc + Inc	8. Inc + Mov
9. Mov + Inc	10. Inc + Nop	11. Nop + Mov	9. Mov + Inc
12. Inc + Add	13. Add + Mov	14. Mov + Nop	15. Nop + Add
16. Add + Bclr			

**Table 5 sensors-23-02917-t005:** CPU0 instruction table and the execution sequence patterns.

Ld	2. Ldi	3. Add	4. Sub	5. Mul
6. Div	7. And	8. Or	9. Xor	10. Shl
11. Shr	12. Cmp	13. Push	14. Pop	15. Mov
16. St				

**Table 6 sensors-23-02917-t006:** CPU0 instruction introduction.

Instructions	Description	Instructions	Description
Ld	Transfer data from memory to register	Xor	XOR two values
Ldi	Transfer immediate data to register	Shl	Shift bits left
Add	Add two values	Shr	Shift bit right
Sub	Subtract two values	Mov	Transfer data between registers
Mul	Multiply two values	Cmp	Compare two values
Div	Divide two values	Push	Store value onto the stack
And	AND two values	Pop	Retrieve the last value from stack
Or	OR two values	St	Transfer data from register to memory (with constant offset)

**Table 7 sensors-23-02917-t007:** Comparison table.

	[10]	2. Platform w/o modified GI	3. Platform w/ modified GI
A. Resolution Δt (ns)	1	0.07	0.07
B. Resolution ratio of A ([10] as 1.0)	1.0	14.28	14.28
C. GI (µs)	60	60	10
D. TestKernel + GI (µs)	65	65	15
E. TestKernel + GI (samples)	65,000	928,571	214,285
F. IOMark number per OSC transmission (count)	16	2	5
G. 1600 IOMark measurements (count)	100	800	320
H. Total measure time (s)	6120	48,120	19,320
I. Measurement time ratio ([10] as 1.0)	1.0	7.86	3.16
J. Manpower (s)	6120	120	120

**Table 8 sensors-23-02917-t008:** Top1 accuracy comparisons between [10] and the proposed platform.

Platform	[10]		Proposed Platform	
NN Model Type	MLP			
DUT	dsPIC	CPU0	dsPIC	CPU0
EM leak	273	273	3898	3898
Validation accuracy	(a) 37.78%	(b) 6.3%~random guessing (1/16 = 6.25%)	(c) 39.24%	(d) 98.71%

## Data Availability

Data is unavailable.

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
