# Peer review of "Improvement of Electromagnetic Side-Channel Information Measurement Platform"

_sensors, 2023, doi:10.3390/s23062917_

Round 1
Reviewer 1 Report
In this paper, a EMR measurement and pattern recognition platform was proposed, which can provide the EMR patterns for neural network (NN) analysis. Also, it improves the measurement flexibility from simple MCUs to field programmable gate array intellectual properties. At last, two DUTs are both tested, and the EMR identification accuracy of MCU is improved.
The existing questions:
(1)The paper attempt to verify the performance of the platform with reference [10], but should be given other comparisons with other literatures.
(2) more type of EM-leaks about instructions should be given.
(3) how the platform will be used with encryption process?
Author Response
The existing questions:
The authors thank for the reviewer's suggestions. The authors try their best to answer the questions and make the paper clearer.
(1)The paper attempt to verify the performance of the platform with reference [10], but should be given other comparisons with other literatures.
R1: The efforts of the work series ([9-10, 12-14]) are generally to build two frameworks: one for data acquisition (methodologies, techniques, platforms, etc.), and the other for data analysis (methodologies, NN model & architectures, generalization of different MCU architecture measurement foundations, etc.) on single type MCU EM-leak problems. However, this paper focuses on the data acquisition platform improvements only, which mostly be-longs to the data acquisition part of the work series. Since the performance comparisons are to be made meaningfully from the whole system (fair comparisons), the proposed platform comparisons must be described based on the integration of the two frameworks with other parts of the two frameworks remaining the same (an ablation study). We can only choose [10] for comparison because it is the State-of-the-art to the best of the authors' knowledge. (Page 2 line 48-57).
(2) more type of EM-leaks about instructions should be given.
R2: We added three new types of EM-leaks in this revision. The instruction added are cmp, push and pop. The instructions are shown in table 6 on page 6. The experiment results are shown in section 4.3, page 14 fig. 18 and table 8. The results are similar to the previous paper. It also shows the scalability of the proposed platform.
(3) how the platform will be used with encryption process?
R3: The authors apologize for their unclear description. It is revised in page 2 lines 67-69. The platform is not used for encryption process. The purpose of the measurement platform is to gather EMR from MCUs in order to reduce the possibility of SCA. However, other researches on the EMR information leakage of specific chips can be benefited from this research

Reviewer 2 Report
The manuscript presents a method to collect and analyze side-channel information
leaked in the form of electromagnetic radiation by microcontrollers (MCU).
The authors measure such radiations with the help of a cartesian robot to automate the
gathering of data. The measurements are performed on a physical MCU and a soft MCU instantiated on to an FPGA.
The topic is certainly interesting in my opinion, but lack in novelty and originality considering the work by the author is essentially derived from [10] ad others.
I noticed a couple of major concerns.
1) In various section of the manuscript (e.g., Abstract) it is stated that "a new platform is proposed".
This is not exact and overall misleading. The authors derive their current work from [10]. It must be stated clearly.
2) The work is *heavily* based on [10]. Very heavily. An author of this manuscript is the author of [10].
In my opinion the authors must be upfront about this and it must be stated clearly that this work is incremental w.r.t. [10], instead of citing continuously ref. [10], especially in Sect. 1 - Introduction (see comment n.3).
3) Several times throughout the manuscript, especially in Sect. 1 - Introduction, are reported comparison with [10].
In view of my comment n.2, I think the authors should write a dedicated section to compare the current work with the previous work in [10].
In such a way, the authors can explicitly highlight the differences (and the similarities).
4) In view of my comments 2 and 3, I believe the introduction should be rewritten to better introduce by briefly describing the topics covered, such as EMR, Side Channel Information, challenges in the measurements, etc.
5) It looks to me that in this manuscript the authors cite and join their previous works in [8], [9], [10], [16], [26], [27]. However, there is no State-of-the-art. There are plenty of studies focusing on EMR for Side Channel. I suggest the authors to expand their view in the State-of-the-art and to limit their citations as well.
6) The author talks about mapping the electromagnetic leakage to a specific instruction. In Table 1-5 are reported the instructions. However there is no mention at all about the data. A MOV instruction could read into main memory and I am pretty sure the EMR is different from that generated by a MOV instruction which does not. I consider this a very important aspect in which the scientific soundness of the manuscript is doubtful.
7) Sect. 3.1: The main characteristics (e.g., resolution, accuracy, ...) of EMR probe and oscilloscope are missing. How long does the robotic arm moves? Which is the path that it follows? Some of these information are mentioned throughout the manuscript but they must be reported here.
Minor comments:
1) Page 5, lines 145 to 150 are very confusing and unclear. Please rephrase.
2) Why Table 2 and Table 5? Please put the code directly into a listing. It is better and more clear.
3) Table 3 and Table 5 could be placed before the respective listing of the code to better understand the meaning of instructions.
4) I noticed in Fig. 5 there are two points where accuracy decreases abruptly. Why are there these "peaks"? (around Epoch 1500 and 2500)
5) Fig. 4 is unclear. Please add an explanation either in the label or in the text-
6) In Fig. 4 ad Fig. 14 the arrows are difficult to see. I suggest another color.
7) Please review with a spell-check
Author Response
The existing questions:
The authors thank for the reviewer's in-depth reviews and suggestions. The authors try their best to answer the questions and make the paper clearer.
1) In various section of the manuscript (e.g., Abstract) it is stated that "a new platform is proposed".
This is not exact and overall misleading. The authors derive their current work from [10]. It must be stated clearly.
R1:
This revision modifies the abstract (lines 16–18) and other sections (see R2) for better descriptions of the contributions.
Yes, the platform is really improved from [10]. For fair comparisons, the platform is really improved from [10]. However, the improvements are non-trivial and heavily modified (for example, time-skew improvements, testKernel identification, control automation for NN model dataset generation, HW/SW-co-integration, target chip extensions). All these improvements make the high resolution data acquisition and time-efficiency possible. Many issues are discussed and com-pared based on [10] in Table 6. This description is added on page 14 lines 400-404 for clarity.
In the authors' viewpoint, the proposed platform indeed makes non-trivial improvements. This description is added on page 15 lines 426.
2) The work is *heavily* based on [10]. Very heavily. An author of this manuscript is the author of [10].
In my opinion the authors must be upfront about this and it must be stated clearly that this work is incremental w.r.t. [10], instead of citing continuously ref. [10], especially in Sect. 1 - Introduction (see comment n.3).
R2: The authors regret for not properly describe the novelty and comparisons of the proposed platform. The authors try their best to clarify these issues addressed by the reviewer. In this revision, they are addressed directly in the abstract (page 1, lines 16-18, page 2 lines 48-57, lines 62-66, page 17 lines 400-404 and 426.
3) Several times throughout the manuscript, especially in Sect. 1 - Introduction, are reported comparison with [10].
In view of my comment n.2, I think the authors should write a dedicated section to compare the current work with the previous work in [10].
In such a way, the authors can explicitly highlight the differences (and the similarities).
R3: The authors thank the reviewer for his in-depth paper review.
As pointed out by the reviewer, the paper is a partial improvement on a series of published works ([9-10, 12-13]), the entire paper describes the concepts, differences, improvements, and compared to the published work series. The authors also try their best to explain in this revision that this paper is different from:
- Other papers (page 2 lines 62-66): The current EM-leak studies generally focus on specific functional chips (DSPs, encryption/decryption chips, among others) [4-7]. While in the work series focus on the general purpose MCU EM-leak issues. And the MCU-EM-leak issues are much harder compare to the specific function chip EM-leak problems. It is because this issue can be generalized to a binary classification problem which is a special case of this research (a multiple classi-fication problem).
- The work series (page 2 lines 48-57): The efforts of the work series ([9-10, 12-14]) are generally to build two frameworks: one for data acquisition (methodologies, techniques, platforms, etc.), and the other for data analysis (methodologies, NN model & architectures, generalization of different MCU architecture measurement foundations, etc.) on single type MCU EM-leak problems. However, this paper focuses on the data acquisition platform improvements only, which mostly be-longs to the data acquisition part of the work series. Since the performance comparisons are to be made meaningfully from the whole system (fair comparisons), the proposed platform comparisons must be described based on the integration of the two frameworks with other parts of the two frameworks remaining the same (an ablation study). We can only choose [10] for comparison because it is the State-of-the-art to the best of the authors' knowledge.
In this revision, new paragraphs (R3) more clearly state the differences and improvements from the previous paper. In section 4.1 Table 7(page 12), the authors have shown the experiment results. The comparisons with [10] are also summarized in the same section.
4) In view of my comments 2 and 3, I believe the introduction should be rewritten to better introduce by briefly describing the topics covered, such as EMR, Side Channel Information, challenges in the measurements, etc.
R4: The authors try to improve this information in this revision as the reviewer suggested. They are substantially modified (page 1 lines 30 - page 3 lines 108).
5) It looks to me that in this manuscript the authors cite and join their previous works in [8], [9], [10], [16], [26], [27]. However, there is no State-of-the-art. There are plenty of studies focusing on EMR for Side Channel. I suggest the authors to expand their view in the State-of-the-art and to limit their citations as well.
R5: It is true that there are plenty of studies focusing on EMR for Side Channel, but there are only a handful of papers that use the platform to automatically measure and distinguish EMR on general purpose MCU as described in R2, R3, and R4. And [10] is also the state-of-the-art to the best of the authors' knowledge. The authors ask for more information from the reviewer about the state-of-the-art works and are happy to include them.
Although the series of papers ([8-10, 12-14]) focus on the MCU's EMR information leakage issues, the detail research topics are different. For example, [8] focus on distinguish EMR by CNN, [9] focus on establishing a measurement platform, [10] focus on distinguish EMR by DNN, [12] focus on time-domain waveform synchronous cutting algorithm and implementation, [13] focus on a new method for creating an ideal square wave based on the synchronous cutting signal so that the synchronous cutting is not affected by noise, [14] focus on a direct extension for program-dependent near-field EMI (rEMI) behaviors under a complex HW/SW environment and overcome the difficulty by finding a size-reduction general rule.
6) The author talks about mapping the electromagnetic leakage to a specific instruction. In Table 1-5 are reported the instructions. However there is no mention at all about the data. A MOV instruction could read into main memory and I am pretty sure the EMR is different from that generated by a MOV instruction which does not. I consider this a very important aspect in which the scientific soundness of the manuscript is doubtful.
R6: The authors understand the reviewer's concern. However, the phenomena are observed and are solved in [Use subsequent neural network to analyze ICEMI, doi: 10.1109/ICCE-TW52618.2021.9602886.] and [Tree Structure Network: A Learning-Based Deep Network for Classification of CPU Instruction through EM Signal, doi: 10.23919/EMCTokyo.2019.8893929] using the NN architecture proposed by the authors. Since it is not the state-of-the-art, the authors brief some techniques below.
Generally speaking, the authors think that the reviewer's intuition is correct but he jumps to the conclusion too soon and miss-judged the paper. The authors have done much efforts in this revision to answer the concern.
First, the reviewer is correct about the EM-leak of instructions groups on memory-access type instructions (MOV and PUSH) and non-memory-access type instructions (ADD and SUB) are different.
The figures below show the time-domain and frequency-domain EM-leaks of 4 instructions including both types. It can be seen that the EM-leaks of the 4 instructions are different. The intuition is correct, at least for the testing targets of the paper.
However, the state-of-the-art already classified (mapped) different instructions' EM-leak data into multiple groups and the binary intuition leads nowhere to the general MCU-EM-leak multiple classification problem. Thus, the authors tried to solve the problem by NN model for multi-group classification and got more accurate results.
Targeting to the reviewer's conclusion, new measurements of dataset for the CPU0 are acquired and analyzed. The figure below is the confusion matrix of CPU0 of both types including MOV. The total Instruction-15 in the figure is MOV. It can be seen that all of the accuracies are high – not just the MOV instruction.
If the reviewer's conclusion is correct that the scientific soundness of the manuscript is doubtful, from the authors' viewpoint, the reviewer implicitly acknowledges that both types should behave very differently to each other. And the reviewer's conclusion means the following conditions should be ALL true:
- The non-diagonal matrix values between types can be cleanly divided (the non-diagonal values between different types will be many zeros/small-values) because the classification between different types of instructions is easy.
- The non-diagonal matrix values within one type cannot be cleanly divided (some/many of the non-diagonal values within the same type will be non-zero/large-values) because the instructions of the same type are too similar to be classified.
However, it is not the case (condition (a) is true but condition (b) is false). This means that there are no such obvious differentiations between the 2 types. Actually, the NN model we proposed can map all the instructions to its own group without any simple/binary classification based on domain knowledge or intuitions. It also means the MOV is not so special comparing to other instructions with respect to the proposed NN model.
So, the reviewer's concern is concluded as:
Q1) The author talks about mapping the electromagnetic leakage to a specific instruction. In Table 1-5 are reported the instructions. However there is no mention at all about the data.
R6-1: Yes, this paper focuses on the measurement platform improvement part of previous work series. From the authors' viewpoint, it is also the main research direction of this journal. And the "no mention at all about the data" is intentionally skipped from obscuring the focus.
Q2) A MOV instruction could read into main memory and I am pretty sure the EMR is different from that generated by a MOV instruction which does not.
R6-2: Yes, as the time/freq-domain waveform shown above, the authors totally agree with the reviewer's opinion on the two testing targets even before this paper was written.
Q3) I consider this a very important aspect in which the scientific soundness of the manuscript is doubtful.
R3: The authors try to convince the reviewer that:
- The authors totally agree with that the review's intuition is sound and the EM-leak differences between types of instructions do exist
- However, even the fact does exist, it leads nowhere for solving the multi-classification/mapping issues. The intuitive binary classification of these two types does not help to classify/map multi-class EM-leak problem.
- The authors already solved it in several previous works through NN. This is why the authors proposed an NN model for analysis – not just because the NN model is a hot topic and try to jump on the bandwagon. And it is not mentioned in this paper to avoid obscuring the focus because it is not related to the work.
- The paper tries to compare the state-of-the-art. And these fundamental works are not the state-of-the-art nowadays. However, the state-of-the-art work inherits from these works.
- It does not mean the soundness intuition on data acquisition justifies the doubts of the scientific soundness of the data analysis framework part of previous works.
7) Sect. 3.1: The main characteristics (e.g., resolution, accuracy, ...) of EMR probe and oscilloscope are missing. How long does the robotic arm moves? Which is the path that it follows? Some of these information are mentioned throughout the manuscript but they must be reported here.
R7:
The authors thank for the reviewer's suggestion. In this review, these parameters are given in page 10 lines 258-259 and 275-277. EMR probe’ s resolution is 5mm. oscilloscope’s bandwidth is 200 MHz to 2GHz, sample rate is 5 G sample/s. It takes 0.5 seconds for the robotic arm to move from point to point. The distance between two points is 2mm. Robotic arm movement path is meandering like the fig below
|
|
||
|
|
||
|
|
|
Minor comments:
1) Page 5, lines 145 to 150 are very confusing and unclear. Please rephrase.
2) Why Table 2 and Table 5? Please put the code directly into a listing. It is better and more clear.
3) Table 3 and Table 5 could be placed before the respective listing of the code to better understand the meaning of instructions.
4) I noticed in Fig. 5 there are two points where accuracy decreases abruptly. Why are there these "peaks"? (around Epoch 1500 and 2500)
5) Fig. 4 is unclear. Please add an explanation either in the label or in the text-
6) In Fig. 4 ad Fig. 14 the arrows are difficult to see. I suggest another color.
7) Please review with a spell-check
R1: It has been modified to describe the pipelining of the MCU at line 154-161. A new table (table 2) is also drawn to give readers a clearer understanding.
R2: We show Table 2 and 5 for different MCU: Table 2 is the instruction description of Table 2. dsPIC33EP512MC202 while the Table 5 is the of CPU0 MCU. They are similar but not exactly the same. From the authors' viewpoint, the separation makes the paper more readable.
R3: The authors thank the reviewer's suggestions. Table 3 and Table 5 are re-placed in this revision as suggested by the reviewer.
R4: The authors do not clearly know about the sudden drop comes from. However, some speculations from the authors are: since the neural network training algorithm we choose (Adaptive Moment Estimation, Adam) changes the learning rate during training in order to find a better fitting. A decrease in the learning rate at some Epochs may cause a sudden accuracy drop in the curve. This phenomenon can be found in other NN training papers, such as: Ibrokhimov, B.; Kang, J.-Y. Deep Learning Model for COVID-19-Infected Pneumonia Diagnosis Using Chest Radiography Images. BioMedInformatics 2022, 2, 654-670. https://doi.org/10.3390/biomedinformatics2040043
R5: Since other operation procedures are the same as [10], only detailed measurement data parameters are included (page 3 lines 137-138).
R6: The authors thank the reviewer's suggestions. They have been changed and much clear in this revision.

Round 2
Reviewer 2 Report
The manuscript presents a method to collect and analyze side-channel information
leaked in the form of electromagnetic radiation by microcontrollers (MCU).
The authors measure such radiations with the help of a cartesian robot to automate the
gathering of data. The measurements are performed on a physical MCU and a soft MCU instantiated on to an FPGA.
This is the second review for this manuscript.
I believe the improvements are significant.
The authors certainly improved the previous version of the manuscript and they worked to correct the previous issues.
Moreover, the authors provide a cover letter that extensively addresses the concerns of the reviewer.
I still consider the present manuscript to be heavily based on a previous work, but the authors are now upfront and the explanations are more clear.